# Swarical: An Integrated Hierarchical Approach to Localizing Flying Light Specks

## ABSTRACT

Swarical, a Swarm-based hierarchical localization technique, enables miniature drones, Flying Light Specks (FLSs), to accurately and efficiently localize and illuminate complex 2D and 3D shapes. Its accuracy depends on the physical hardware (sensors) of FLSs, which are used to track neighboring FLSs in order to localize themselves. It uses the hardware specification to convert mesh files into point clouds that enable a swarm of FLSs to localize at the highest accuracy afforded by their hardware. Swarical considers a heterogeneous mix of FLSs with different orientations for their tracking sensors, ensuring a line of sight between a localizing FLS and its anchor FLS. We present an implementation using Raspberry cameras and ArUco markers. A comparison of Swarical with a state of the art decentralized localization technique shows that it is as accurate and more than 2x faster.

Click ISR, HC, and RSF for anonymized video links of a demonstration of Swarical's localization techniques. See anonymized video links for a comparison of SwarMer and Swarical.

**ACM Reference Format:**
. 2018. Swarical: An Integrated Hierarchical Approach to Localizing Flying Light Specks. In *Woodstock '18: ACM Symposium on Neural Gaze Detection, June 03–05, 2018, Woodstock, NY*. ACM, New York, NY, USA, 9 pages. https://doi.org/XXXXXXX.XXXXXXX

## 1 INTRODUCTION

A Flying Light Speck (FLS) is a drone configured with light sources [14]. A swarm of FLSs may illuminate complex 2D and 3D multimedia shapes in a fixed volume, a 3D multimedia display [15]. Each FLS is assigned a coordinate. A challenge is how cooperating FLSs may illuminate 2D and 3D shapes. Use of GPS [26] is not an option due to the lack of a line of sight to GPS satellites in an indoor setting [3, 14]. An FLS may travel to its assigned coordinate using Dead Reackoning [6]. This technique may employ a drone's inertial measurement unit (IMU) to approximate its location. IMUs of a drone are known to be noisy, with the error in estimated location increasing as a function of traveled distance [3, 6, 17, 22]. Figure 1 shows a palm tree with different degrees of dead reckoning error.

A localization framework may manipulate a design space consisting of hardware, software, and data. Consider each in turn: Hardware includes sensory devices mounted on an FLS. A framework has a host of hardware choices ranging from Ultra Wide Band

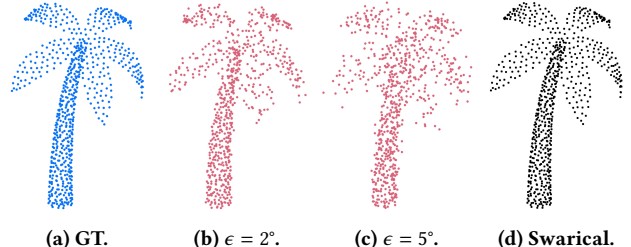

**(a) GT.**  **(b)** $\epsilon = 2°$.  **(c)** $\epsilon = 5°$.  **(d) Swarical.**

**Figure 1: Palm tree with 725 FLSs. Ground truth (GT), dead reckoning with two different degrees of error ($\epsilon = 2°$ and $5°$), and Swarical using dead reckoning with $\epsilon = 5°$.**

(UWB) radios [10, 27, 28] to ultrasonic devices and cameras [20, 23–25]. The software includes algorithms that implement a localization technique. A framework may use the decentralized algorithm of SwarMer [3] that is executed by FLSs. Data refers to a 3D shape and its representation as a point cloud. An example of a 3D shape file is a polygon mesh file. It is a collection of vertices, edges, and faces that define a 3D shape. A framework may adjust the number of FLSs used to illuminate the faces of a mesh file. With different types of FLS hardware, the framework may use a mix of FLSs that enhance the accuracy of localization, which enables a swarm of drones to illuminate a shape with high accuracy.

In this paper, we present a Swarm-based hierarchical (Swarical) framework to localize FLS. Swarical is an integrated approach that considers hardware, software, and data to localize FLSs. It starts by selecting the hardware that enables FLSs to localize. It uses the specification of this hardware in combination with a mesh file to compute the number of FLSs that should illuminate the shape. This considers the range of sensors used to localize FLSs in combination with the characteristics of a mesh file. Given a heterogeneous mix of FLSs with different mountings of sensors (for a line of sight), Swarical computes the right mix of FLSs to illuminate a shape. This mix ensures a localizing FLS has a line of sight with its anchor FLS.

**Contributions** of this paper include:

- Swarical as a framework that considers hardware, software, and characteristics of a mesh file (data) to compute a point cloud for localization and illumination of a shape. (Sections 2 and 3.)
- Three test localization techniques with ISR emerging as the superior technique, offering enhanced speed and accuracy compared to its counterparts. (Section 4.)
- An implementation of Swarcial using cameras and ArUco markers mounted on FLSs to track one another. (Section 5.)
- A comparison of Swarical with a state of the art decentralized algorithm named SwarMer [3] shows Swarical is more than 2x faster and equally accurate. (Section 5.4.)

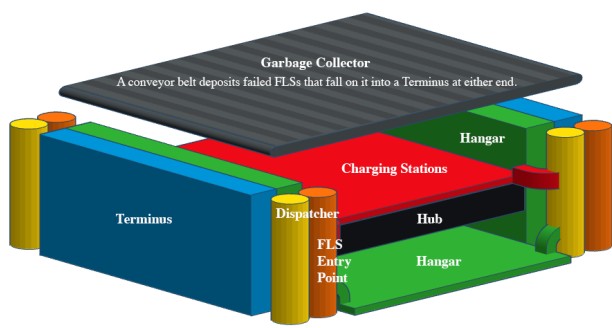

**Figure 2: The yellow cylinders of the architecture of [15] are dispatchers that deploy FLSs. The Hub is comparable to today's servers and hosts the Orchestrator process.**

- We open source our software implementations and its data set at https://anonymized.

**Related work:** The concept of 3D displays using FLS is presented in [1–5, 8, 9, 14–16, 23, 31, 32]. The most relevant is SwarMer, a decentralized localization technique that is fast and highly accurate. A qualitative and quantiative comparison of SwarMer with Swarical is presented in Section 5.4. Obtained results show Swarical is equally accurate and more than 2x faster than SwarMer.

The rest of this paper is organized as follows. Section 2 provides an overview of Swarical and establishes the terminology used in this paper. While Section 3 introduces the planner component of Swarical, Section 4 introduces several online decentralized localization techniques. We introduce an implementation of Swarical in Section 5 and compare it with SwarMer [3]. Brief conclusions are presented in Section 6.

## 2 OVERVIEW AND TERMINOLOGY

This paper assumes the architecture of [1, 15], see Figure 2. It consists of a hub and one or more dispatchers to deploy FLSs. The Hub is a computer similar to today's servers. It hosts an Orchestrator process that executes the planner component of Swarical, see Figure 4. The Orchestrator provides metadata to FLSs and deploys them using one or more dispatchers. The FLSs travel to their assigned coordinates using Dead Reckoning. They localize relative to one another to illuminate 2D and 3D shapes.

An FLS may be configured with various sensors that enable it to localize relative to a neighboring FLS. Section 5.1 describes the use of cameras and ArUco markers [13]. A localizing FLS uses its camera to take a picture of its anchor FLS's ArUco marker and processes the picture to compute its relative pose to the anchor FLS. A challenge is how to mount cameras and ArUco markers on FLSs to ensure the camera of a localizing FLS has a line of sight with the ArUco marker of its anchor FLS. We address this challenge using a heterogeneous mix of FLSs with cameras mounted on their top, side, or bottom. See Figure 3.

Swarical is a divide-and-conquer technique. It partitions a shape into a collection of swarms. FLSs of a swarm localize relative to

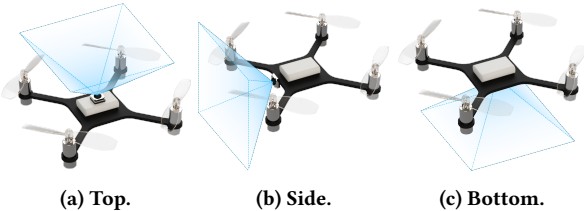

**(a) Top.**     **(b) Side.**     **(c) Bottom.**

**Figure 3: Three FLSs with different camera orientations/FoVs.**

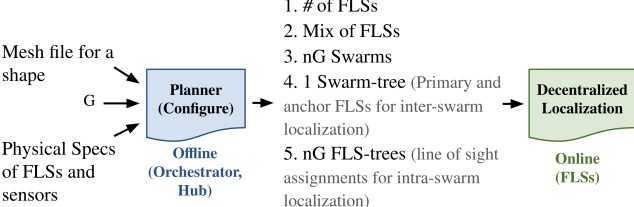

**Figure 4: Swarical, a divide-and-conquer framework.**

one another. This is intra-swarm localization. A swarm also localizes relative to another swarm. This is inter-swarm localization, stitching swarms together to illuminate a complex 2D/3D shape.

Swarical consists of two distinct steps, see Figure 4. A centralized configuration planner and a decentralized localization process. The former is an offline process executed by the Orchestrator. The latter is an online technique executed by swarms of FLSs.

The input to the planner is a mesh file of a shape, the desired size of a swarm (G), and the available mix of FLSs with the specification of their sensors (e.g., range of a sensor). The planner processes the mesh file to compute both the number of FLSs and their correct mix to illuminate the shape using the specification of the localization device. It constructs groups of FLSs that are in close proximity to one another. The size of each group is approximately $G$.

The planner constructs two types of trees, FLS-trees and a swarm-tree. See Figure 5. An FLS-tree defines the anchor FLS for a localizing FLS in a swarm. The swarm-tree identifies a *primary* FLS in a child swarm that localizes relative to an anchor FLS in its parent swarm. The root of the swarm-tree is an exception. Both trees guarantee a localizing FLS has a line of sight with its anchor FLS.

When illuminating a shape, FLSs that constitute a swarm continuously localize relative to one another. The primary of a swarm (except for the root) will localize relative to the identified anchor FLS of its parent swarm. It computes a vector for its movement. Its entire swarm, including the primary, moves along this vector.

DEFINITION 1. *A swarm consists of one or more FLSs. Members of a swarm localize relative to one another continuously. A swarm-tree identifies the parent-child relationships between swarms. Except for the swarm that serves as the root of the tree structure, every swarm has a parent swarm and one FLS $f_P$ designated as its primary. The primary $f_P$ of a child swarm localizes relative to an anchor FLS of its parent swarm, computing a vector $\overrightarrow{V}$. $f_P$ and all FLSs that constitute its swarm move along this vector $\overrightarrow{V}$.*

The output of the planner may be a large volume of data. However, each FLS requires a small fraction of this output to cooperate with the other FLSs by executing the decentralized localization technique. (See Section 4.1 for details.) The Orchestrator provides this information to the individual FLSs.

For a given shape, the Orchestrator may execute the planner and store its output in a file. When a user requests the display of the shape repeatedly, the Orchestrator may read the file to provide each FLS with the required information [4]. The FLS localization process is decentralized, fast, and continuous.

## 3 PLANNER

The planner consists of two sequential steps. First, it converts a mesh file into an FLS point cloud using the limits of a tracking device. Second, it fragments the resulting point cloud consisting of $F$ FLSs into $nG$ swarms. Each swarm consists of approximately $G$ FLSs. This step constructs one swarm-tree and $nG$ FLS-trees, $nG = \lceil \frac{F}{G} \rceil$ swarms. Below, we describe the two steps in turn.

### 3.1 Step 1: Mesh File to FLS Illumination

FLSs must track one another to localize in order to illuminate a mesh file. The limits of the FLS tracking device in combination with the error tolerated by an application dictate the number of FLSs used to illuminate a face. To illustrate, consider an application that tolerates 5% error in the maximum difference between the estimated truth and the ground truth, i.e., Hausdorff distance [18]. The application uses the minimum and maximum range ($[T_{min}\text{-}T_{max}]$) of the tracking device that produces at most 5% error in measured distances to compute the density of FLSs in a face. Below, we present a general technique for computing this density. An implementation of it in the context of visual tracking using fiducial markers is presented in Section 5.

Consider a tracking device placed at the center of a spherical shaped FLS with a radius of R. An application tolerates e% error in the Hausdorff distance of an illumination. The planner identifies the minimum and maximum $[T_{min}\text{-}T_{max}]$ range of the tracking device with a percentage error less than or equal to e. Assume the radius R is less than or equal to $T_{max}$, $R \leq T_{max}$, the planner computes the min/max density of FLSs in a unit of area: $D_{min} = \frac{1}{\pi \times max(T_{max}/2,R)^2}$, $D_{max} = \frac{1}{\pi \times max(T_{min}/2,R)^2}$. By multiplying these by the area of a face, the planner estimates the minimum and maximum number of FLSs required to illuminate the face with e% error in Hausdorff distance.

There is extensive work on sampling a mesh file [29] to generate a point cloud. Section 5.2 uses the Constrained Poisson-disk sampling [11] by providing it with the number of FLSs computed using the above discussion. It is possible to use other techniques [30].

### 3.2 Step 2: FLS-Tree and Swarm-Tree

The planner constructs swarms with different mixes of FLSs to facilitate intra and inter swarm localization. Given a group size $G$ and $F$ FLSs, the planner constructs $nG$ groups using the k-Means [21] algorithm, $nG = \frac{F}{G}$. Each resulting group will consist of approximately G FLSs. A group corresponds to a swarm.

The planner constructs a *swarm-tree* on the $nG$ swarms, identifying one FLS of a swarm as its primary $f_P$ that localizes relative to

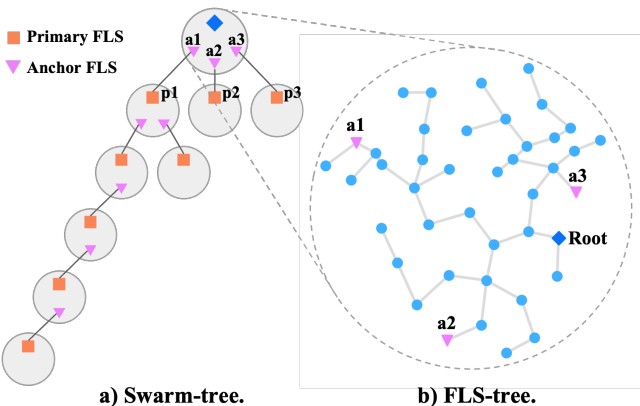

a) Swarm-tree.  b) FLS-tree.

**Figure 5: Swarm-tree and FLS-tree with the Chess Piece, $G$=50.**

the nearest anchor FLS in a parent swarm. The planner constructs an *FLS-tree* on the $G$ FLSs in a swarm, establishing the localizing and anchor relationship between the FLSs that constitute a swarm. Figure 5 shows the FLS-tree and swarm-tree constructed on the Chess Piece.

When constructing either the FLS-tree or the swarm-tree, the objective of the planner is to satisfy two constraints. First, the tracking device of a child FLS should have a line of sight with its parent FLS. Second, the distance between the localizing FLS and its anchor FLS respects the $[T_{min} - T_{max}]$ range of the tracking device.

To realize its objective, the planner uses the center of a swarm to construct a minimum-spanning tree [7, 19] across the swarms. This is the swarm-tree. Its vertices correspond to swarms of FLSs. The weight of an edge between two swarms is the Euclidean distance between the center of the swarms. The minimum spanning tree connects all the swarms together without any cycles and with the minimum possible total edge weight. The planner identifies the vertex with the highest number of edges as the root of the swarm-tree. It walks its children in a breadth first manner to establish the parent-child relationship between swarms. With a parent-child swarm, the planner selects an FLS from the parent swarm that is closest to an FLS in the child swarm. The latter is the primary FLS of the child swarm. It localizes relative to the anchor FLS identified in the parent swarm. The planner uses the orientation of the primary FLS in the point cloud to assign it one of the FLS types in Figure 3 with the objective of ensuring the primary has a line of sight to its anchor.

Once the primary FLS of a swarm is identified, the planner computes a minimum spanning tree for the FLSs that constitute a swarm. This is the FLS-tree. Its vertices correspond to FLSs. The distance between two FLSs is computed using the Euclidean distance between their coordinates. The minimum spanning tree connects all the FLSs together without any cycles and with the minimum possible total edge weight. The planner traverses this tree starting with the primary in a breadth first manner. It establishes the line of sight relationship from child to parent. The planner uses the orientation of an FLS in the point cloud to assign the child FLS one of the FLS types shown in Figure 3. In selecting the type, it ensures a localizing (child) FLS has a line of sight with its anchor (parent) FLS.

With both the swarm-tree and $nG$ FLS-trees, the planner ensures the distance between a localizing and anchor FLS is lower than $T_{max}$. If their distance exceeds $T_{max}$, the planner inserts dark FLSs to reduce the distance. These FLSs may be used as hot standbys to tolerate the failure of the illuminating FLSs [5].

# 4 CONTINUOUS LOCALIZATION

This section describes three localization techniques. All three assume an Orchestrator that allocates the correct mix of FLSs per output of the planner. The Orchestrator uses the FLS-trees and the swarm-tree to assign each FLS a coordinate in the 3D volume and provide it with its parent FLS and children FLSs. With an FLS designated as the primary of a swarm, $f_P$, the Orchestrator provides the FLS with the identity of its anchor FLS in a different swarm (as computed by the planner).

The key difference between the localization techniques is the amount of concurrent movement by different FLSs in a swarm and across the swarms. We start with a highly concurrent technique. Subsequently, we describe two variants that limit the amount of concurrent movement. Our experimental results show the second technique, ISR, is faster and more accurate than the other two. It is also more energy efficient by minimizing the total distance traveled by FLSs.

**Highly Concurrent, HC,** allows the primary of a swarm ($f_P$) to localize relative to its anchor in the parent swarm while the anchor is localizing itself. This means all swarms may localize at the same time. Below, we describe how FLSs in a swarm localize relative to one another, intra-swarm localization. Subsequently, we describe how two swarms localize relative to one another, inter-swarm localization.

Using the ground truth, an FLS knows its position and orientation relative to its swarm members. The FLS-tree ensures a localizing FLS has a line of sight with its anchor FLS. The root of the tree is an exception. Consider localization for a child FLS and a root FLS in turn.

A child FLS $u$ computes its pose relative to its parent $v$, $r_{uv}$ ($r_{uv} = -r_{vu}$). It broadcasts $r_{uv}$ to all its swarm members. A receiving FLS constructs an intra-swarm tree to maintain this information broadcasted by different FLSs. See the FLS-tree of Figure 5. An FLS $i$ computes a relative pose for each reachable[1] FLS within the tree structure. This relative pose, denoted as $R_{ij}$, is determined by the sum of relative pose vectors $r_{uv}$ along the path from FLS $i$ to FLS $j$. To correct its position relative to these FLSs, FLS $i$ computes a correction vector $v_{ij}$, defined as $(P_i - P_j) - R_{ij}$, where $P_i$ and $P_j$ represent the ground truth positions of FLSs $i$ and $j$ respectively. This process is repeated for all reachable FLSs, resulting in a set of correction vectors. FLS $i$ then moves along the average of these vectors, computed as $\frac{1}{N} \sum_{j \in N_T} v_{ij}$, where $N_T$ is the reachable FLSs in the FLS-tree, including FLS $i$. It is possible for an FLS to compute a vector using only its parent FLS. This happens at the very beginning prior to the FLS receiving a vector from other FLSs or when a swarm consists of only 2 FLSs.

---

[1]Reachable means there is a path between the FLS and other FLSs in the tree with information about their relative pose. Either the Orchestrator may provide an FLS with the FLS-tree, or the network transmission of an FLS may include its id and its parent id to enable a receiving FLS to construct the FLS-tree.

Every time an FLS receives the relative pose from another FLS in its swarm, it repeats the process to localize itself. Should an FLS not receive information from its swarm members for 500 milliseconds, it localizes, computes a vector, broadcasts its pose relative to its parent to all its swarm members, and moves along the vector. An FLS clears its tree structure after each inter-swarm localization.

The root FLS also receives relative measurements from its children, grandchildren, great grand children, and other descendent FLSs in the tree. It uses this information to compute its relative pose to them. It computes a vector to correct its position relative to each FLS. Next, it computes an average of these vectors. And, moves along this average vector to localize.

An inter-swarm localization occurs once the length of the vector computed by all members of a swarm is smaller than a pre-specified threshold. Once the primary $f_P$ of a swarm detects this condition, it localizes relative to its anchor in its parent swarm. The root swarm is an exception as it has no primary and will not localize relative to another swarm. $f_P$ uses its pose relative to its anchor to compute a vector to correct its pose. Subsequently, $f_P$ and its entire swarm moves along this vector. After this movement, the FLSs that constitute the swarm clear their tree structure of the relative pose information broadcasted by the FLSs in their swarm. Subsequently, they repeat their intra-swarm localization.

HC prevents a swarm from performing inter-swarm localization while its FLSs are localizing actively, i.e., their computed average vector is greater than a pre-specified threshold. Removing this requirement results in a variant with higher concurrency. It causes FLSs that constitute a swarm to move away from their primary, producing distorted shapes.

**Inter-Swarm Rounds, ISR,** limits the number of swarms that localize at a time. It requires the anchor FLS of $f_P$ to be stationary prior to $f_P$ localizing relative to it. It realizes this objective using the swarm-tree as follows. Once the length of the correction vector computed by an FLS in the root swarm that serves as an anchor for a child swarm, the anchor informs its $f_P$ to localize. The $f_P$ waits until its correction vector relative to its swarm members is smaller than a pre-specified threshold. Subsequently, the $f_P$ localizes relative to its parent's anchor FLS, computes a vector, moves along this vector, and requires its entire swarm to move along this vector. Next, the anchor FLSs in the $f_P$'s swarm notify their children's $f_P$ to localize relative to them. This process continues until the children swarms at the leaves of the tree localize.

ISR's localization is continuous starting with the root swarm. An anchor FLS in one swarm may send multiple notifications to its $f_P$ to localize while the $f_P$ is waiting for its correction vector to become smaller than the pre-specified threshold. In this case, the $f_P$ drops the repeated messages. It localizes once after its correction vector is smaller than the pre-specified threshold.

The root swarm initiates the above process every time it receives a relative pose from a swarm member that causes it to compute a correction smaller than the pre-specified threshold. The concept of a swarm member localizing every 500 millisecond is present. Hence, in the worst case scenario, the root swarm initiates localization every 500 milliseconds.

**Rounds across the Swarm-tree and FLS-trees, RSF,** constrains the number of concurrent localizations within a swarm. An FLS in

a swarm localizes relative to its anchor in rounds. These rounds are initiated by the root of the FLS-tree.

RSF is continuous similar to the other techniques. Starting with the root swarm of the swarm-tree, the root FLS of its FLS-tree notifies each of its children FLSs to localize relative to it while it remains stationary. Subsequently, each child FLS notifies each of its children FLSs to localize relative to it while it remains stationary. This process repeats continuously.

Except for the swarms that are at the leaves of the swarm tree, a swarm has an anchor FLS for each of its children swarms. An $f_P$ of these swarms localizes relative to their anchor. Once an anchor FLS completes its localization, it notifies the $f_P$ to localize relative to it. This causes the entire child swarm containing $f_P$ to move. Subsequently, $f_P$'s children localize relative to it. This process repeats continuously. The root swarm initiates the above process similar to ISR.

## 4.1 Space Complexity

An FLS has a limited amount of memory. While the space complexity of RSF is in the order of hundreds of bytes, the space complexity of HC and ISR is in the order of kilobytes. Below, we present space complexity of each technique and quantify it using our implementation of Section 5.

The amount of memory required from an FLS by RSF is O(M) where M is the size of the metadata provided by the Orchestrator (see the next graph for details). In addition to the metadata, HC and ISR require an FLS to construct an in-memory data structure representing the FLS-tree. This data structure consist of $G$ nodes and $G-1$ edges. Each node maintains the ground truths coordinates (12 bytes) and the id of the FLS (4 bytes) that it represents. Each edge has a parent id (4 bytes), child id (4 bytes), and a relative pose. The latter is a 3D coordinate (12 bytes). Hence, the space complexity of HC and ISR is O(M+G+G-1). In our implementation, it is M+G×16+(G-1)×20 bytes.

The metadata provided by the Orchestrator and maintained by each FLS includes its FLS id (4 bytes), a swarm-id (4 bytes), ground truth coordinates (12 bytes), 1 parent id (4 bytes), and a list of its $C$ children ids (C × 4 bytes). The FLS designated as primary maintains the swarm-id of its anchor FLS (4 bytes). Each anchor FLS in a swarm is provided with the identity of the primary that localizes relative to it (4 bytes). The space complexity of M is O($\epsilon$+C) where $\epsilon$ is a constant. In our implementation, $\epsilon$=32 bytes. Hence, the required memory for M is 32+C×4 bytes. See Figure 12 for the branching factors (C) in our conducted experiments.

## 5 AN IMPLEMENTATION AND EVALUATION

This section describes an implementation of Swarical using Raspberry cameras and ArUco markers. Section 5.1 presents a camera and characterizes its accuracy in measuring post. Subsequently, Sections 5.2 and 5.3 present results from Swarical's planner and localization techniques, respectively. Finally, we compare Swarical with a state of the art decentralized localization technique named SwarMer [3] in Section 5.4.

## 5.1 FLS Tracking: Calibration

To localize relative to its neighbors, an FLS must track them. The ideal tracking mechanism should be:

- **Accurate:** An FLS should be able to measure its state relative to a neighbor. The relative state includes relative position and orientation. Ideally, the accuracy of the position should be in millimeters. The error in a measured orientation should be less than 1 degree in each dimension.
- **Acceptable range:** An FLS should be able to measure its state relative to a neighbor that is a part a few centimeters up to tens of centimeters.
- **Fast with a high refresh rate:** An FLS should be able to quantify its relative state to a neighboring FLS in sub-milliseconds. Moreover, it should be able to refresh this information quickly at a frequency of 10 Hz.
- **Robust:** An FLS should be able to track a neighbor in an indoor setting with different lighting. In a pitch-dark room, an FLS should be able to track its neighbors.

**Table 1: Raspberry camera module 3 NoIR specifications.**

| Lens | Resolution (px) | FoV (°) | Min Focus Range | Weight (g) | Price (USD) |
|------|-----------------|---------|-----------------|------------|-------------|
| Regular | 4608 × 2592 | D 75, H 66, V 41 | 100 mm | 3.2 | $25 |
| Wide | 4608 × 2592 | D 120, H 102, V 67 | 50 mm | 3.2 | $35 |

ArUco markers [13] with a Raspberry camera configured with IR lighting satisfy the above requirements. The camera is small, lightweight, and ready for use with a drone. It has a regular and a wide lens with a minimum focus range of 5 and 10 cm, respectively. (See Table 1.) It supports three different resolutions. Table 2 shows these and our experimentally measured average camera delay and processing time. The average camera delay is the elapsed time from when the application requests a frame to the time the camera provides the frame. Processing time is the time required to measure position and orientation using Raspberry Pi 5. We designed our software to capture an image once it is done processing the current image. Hence, the reported accuracy is based on the latest image available. We use the 720p frame resolution for the rest of this paper.

The maximum range of each camera for detecting a marker depends on the size of the marker. Figure 6 shows the detection rate with a 4.7 mm paper printed maker size. While the x-axis of this figure is the distance between the camera and the marker, the y-axis is the detection rate. It highlights the minimum focus range of Table 1 with the detection rate becoming 100% at those minimums. With the wide angle lens, the detection rate drops to zero with 300

**Table 2: Camera's frame rate and marker detection performance with the regular/wide lens.**

| Resolution | Frames/ Second | Avg Camera Delay, milliseconds | Avg Processing Time, milliseconds |
|------------|----------------|-------------------------------|-----------------------------------|
| 480p | 59.3/44.8 | 10/15 | 6/7 |
| 720p | 46.9/44.4 | 3/8 | 18/14 |
| 1080p | 21.1/26.0 | 8/8 | 39/29 |

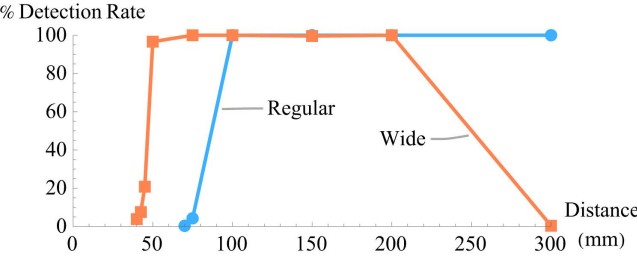

**Figure 6: Detection rate as a function of distance between camera and marker. The paper printed marker size is 4.7 mm.**

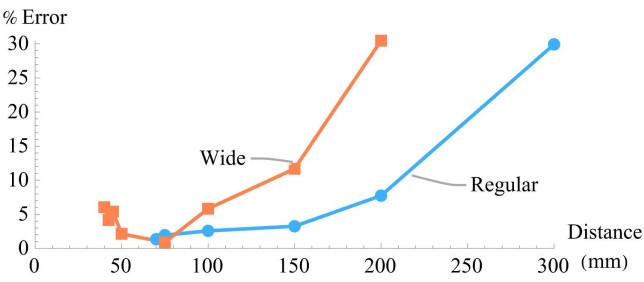

**Figure 7: Percentage error of distance measurement as a function of distance between camera and marker. The paper printed marker size is 4.7 mm.**

mm. With this marker size, Figure 7 shows the percentage error increases as a function of the distance[2]. The regular lens provides a lower error as a function of longer distances when compared with the wide lens.

Larger marker sizes reduce the error with both lenses. The wide lens has a lower error when compared with the regular lens. See Figure 8. In this figure, the x-axis is the marker size, and the y-axis is the percentage error in the measured distance. The reported errors are for measuring the minimum focus range of the two lenses, 5 and 10 cm, with wide and regular lenses, respectively. In general, paper provides a higher percentage error when compared with LCD.

Figure 9 shows the error in roll, pitch, and yaw as a function of paper printed marker size with the wide lens. The camera provides a higher accuracy for the yaw (rotation around the axis perpendicular to the marker) than the roll (rotation around the length) and the pitch (rotation around the depth). This accuracy decreases with marker sizes smaller than 3 mm.

In darkness, an FLS may use the camera with IR light to capture an image of paper-printed markers for processing. In our experiments, IR lighting in the dark does not impact the accuracy of measurements and the detection rate.

## 5.2 Planner

We use the range of the Raspberry camera at [6,8] cm as it provides the highest accuracy. With this range, the planner computes a point cloud of 1855 FLSs for the skateboard. The mix of FLSs with a

---

[2]With both lenses, we report the percentage error with distances smaller than the advertised minimum as long as the camera detects the ArUco marker.

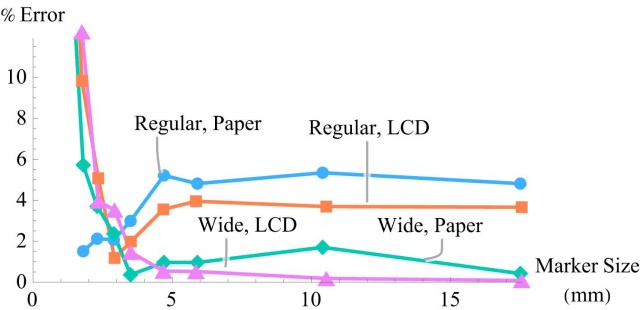

**Figure 8: Percentage error of distance measurement as a function of marker size. The measured distance is 5 and 10 cm for wide and regular lens, respectively.**

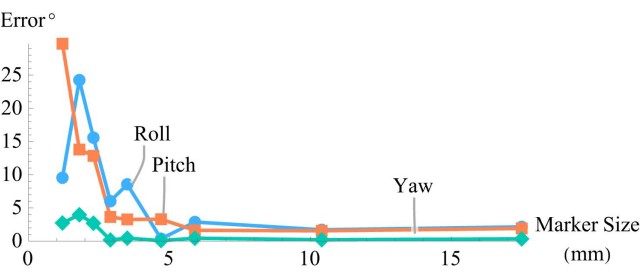

**Figure 9: Error of orientation measurements in degrees as a function of marker size with wide lens and printed markers.**

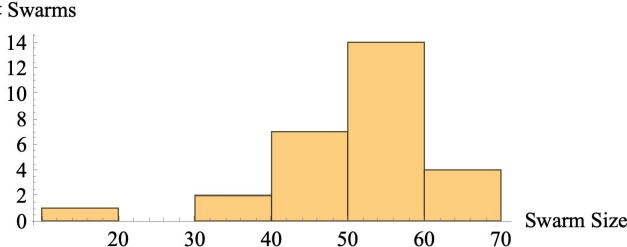

**Figure 10: Distribution of swarm size, Skateboard, *G*=50.**

camera mounted on their top, side, and bottom is 152, 1585, and 118, respectively. The percentage of each variant is 8.2%, 85.4%, and 6.4%, respectively. This mix ensures a localizing FLS has line of sight with the ArUco marker of its anchor FLS. With all the shapes, the percentage of FLSs with a camera mounted on their side is significantly higher than the others.

Figure 10 shows the distribution of swarm size with the Skateboard with *G*=50. Swarical uses k-Means to construct swarms. This clustering technique minimizes the euclidean distance between the FLSs that constitute a swarm. However, it does not ensure swarms of the size. As shown, the size of a swarm varies from 10 to 90. The same is true with the other shapes. The topology of a shape dictates the swarm sizes constructed by k-means.

Figure 11 shows the distribution of distance between localizing and anchor FLSs within the swarms (FLS-trees) and across swarms

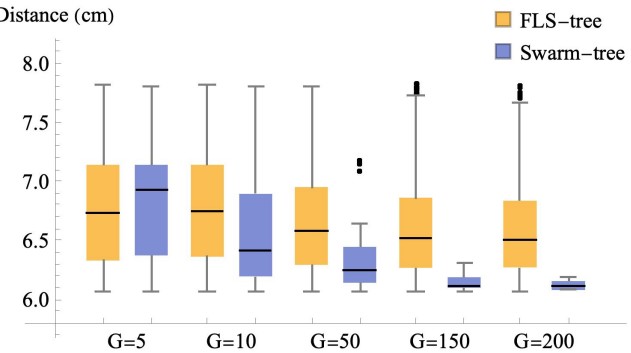

**Figure 11: Distribution of distance between localizing and anchor FLSs within a swarm (FLS-tree) and across swarms (Swarm-tree), Skateboard.**

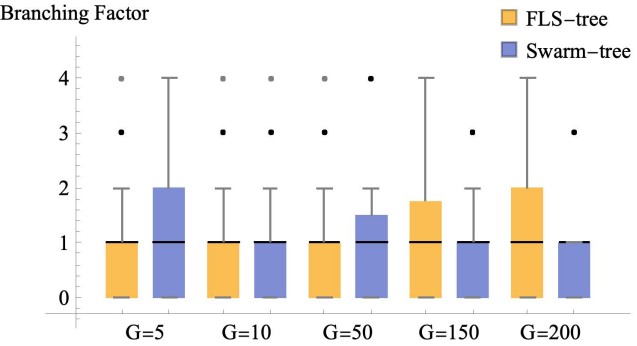

**Figure 12: Distribution of the number of localizing FLSs (swarms) per anchor FLS, Skateboard.**

(swarm-tree). This is for the skateboard with different group sizes, $G$. In these experiments, the planner was configured to limit the distance between a localizing and an anchor FLS to [6-8] cm. After constructing the swarm-tree and FLS-trees for each value of $G$, it inserted a total of 160, 132, 40, 23 and 18 dark FLSs for $G$=5, 10, 50, 150, and 200, respectively. Hence, the median is 6 and 7 cm for all $G$ values. There is no localizing-anchor pair with a distance smaller than 6 cm. The variation in distance is greater for the swarm-tree with smaller group sizes, $G$=5 and 10. This is because there is a larger number of swarms. The inverse is observed with larger group sizes, $G$=150 and 200, because there are fewer swarms.

Figure 12 shows the distribution of the branching factor for the swarm-tree and the FLS-trees. This is the number of FLSs (swarms) that localize relative to one anchor FLS (swarm). The median is one. However, the outliers may be as high as 3 or 4. The minimum is zero. These correspond to FLSs (swarms) that are the leaves of an FLS-tree (swarm-tree).

## 5.3 Localization

All experiments reported in this section are conducted using a cluster of 20 Amazon AWS servers, c6a.metal with 192 virtual cores. Each core is used to emulate an FLS. We use Hausdorff Distance (HD) [18] and Chamfer Distance (CD) [12] to compare the quality of

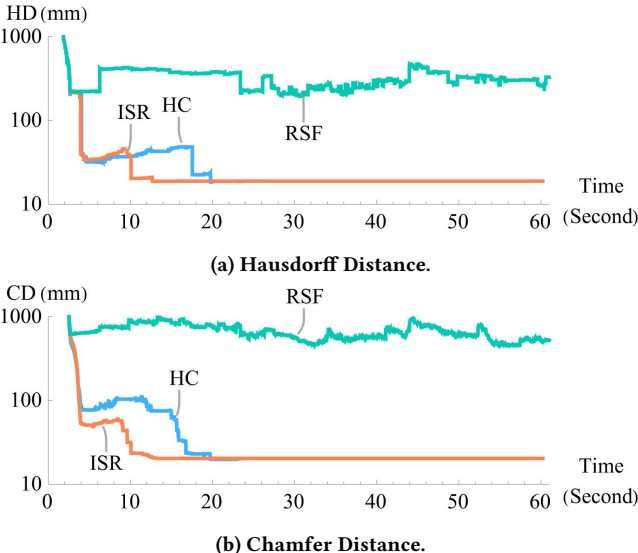

**(a) Hausdorff Distance.**

**(b) Chamfer Distance.**

**Figure 13: A comparison of Localization techniques, Skateboard, $G = 50$. Click ISR, HC, and RSF anonymized video links for a demonstration.**

localizations provided by HC, ISR, and RSF. These mertics compare the FLS coordinates obtained using a localization technique, i.e., the estimated truth $E$, with the FLS coordinates provided by the Planner, i.e., the ground truth $P$. HD quantifies the maximum error in distance between $E$ and $P$ after applying a translation. CD quantifies the average error between $E$ and $P$. Both techniques require a translation process because Swarical is a relative localization technique. Our implementation of the translation process computes the center of $E$ and $P$. It aligns their centers prior to measuring the maximum/average error. A lower value is better with zero reflecting a perfect match between $E$ and $P$.

In general, HD is more strict that CF because it uses the maximum error. Both are useful in understanding the tradeoffs associated with the alternative techniques.

Figure 13 compares HC, ISR and RSF with one another. The x-axis is the elapsed time from when the dispatcher deploys the first FLS. Once an FLS arrives at its assigned coordinate, it starts to localize. We assume a 5° dead reckoning error. The y-axis shows the HD and the CD[3] in Figure 13a and 13b, respectively. Both figures are for the Skateboard with $G$=50. Similar trends are observed with the other shapes and values of $G$.

Figure 13 shows ISR is superior to HC and RSF. It enhances both HD and CD, providing illuminations that resemble those computed by the Planner more accurately. RSF is significantly worse. It requires an FLS to to compute its pose relative to another FLS (its anchor). HC and ISR require an FLS to compute an average correction pose. This averaging minimizes HD and CD as a function of time while RSF's HD and CD remain elevated.

---

[3]While CD is an averaging, it is possible for its computed value to be higher than HD. It computes the average distance between point clouds A and B, then computes it again by replacing A with B, for B and A, and then adds the two values [12]. See Equation: $\text{Chamfer}(A, B) = \frac{1}{|A|} \sum_{a \in A} \min_{b \in B} \|a - b\|_2^2 + \frac{1}{|B|} \sum_{b \in B} \min_{a \in A} \|b - a\|_2^2$.

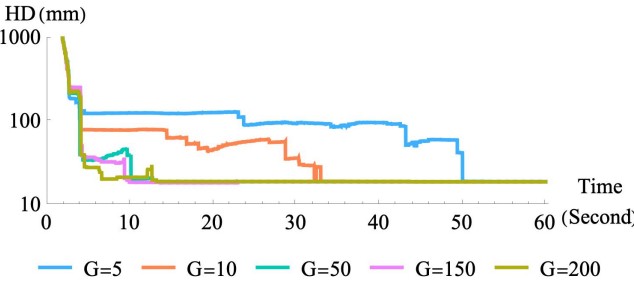

(a) Hausdorff Distance.

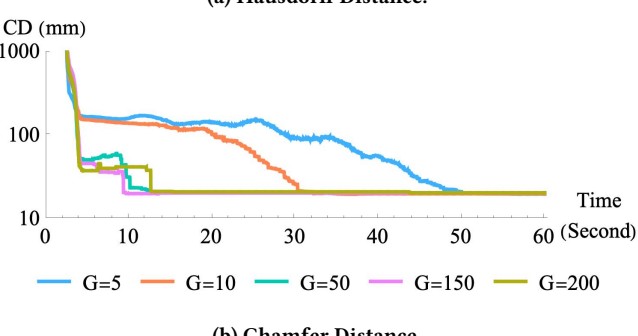

(b) Chamfer Distance.

**Figure 14: Comparison of different swarm sizes ($G$) with the Skateboard and the ISR technique. Lower is better.**

In all these experiments, RSF causes the FLSs to travel a longer total distance when compared with ISR and HC. ISR reduces this metric slightly lower than HC. This slight improvement is consistent throughout our experiments.

The select range of [6,8] cm corresponds to 0.9 to 1.2 mm error, see Figure 7. However, in Figure 13, HD levels off at 18.9 mm. This is 20x higher. If we considered only two points then we would observe the expected 0.9 to 1.2 mm error. However, with a point cloud, the error compounds as FLSs localize to magnify the error.

Figure 14 shows the HD and CD of the Skateboard with ISR and different swarm sizes ($G$). Small swarm sizes ($G \le 10$) result in a higher HD and CD, i.e., a larger difference between the FLS clouds illuminated by ISR and the FLS cloud computed by the Swarical's planner. This is because they result in a larger number of swarms that is unbalanced and deep, 43 with $G$=5 and 38 with[4] $G$=10. The swarms close to the leaves of the swarm-tree require a longer time to localize because their anchor in a parent swarm has a higher probability of changing its location. This change is due to both intra-swarm and inter-swarm localization. An inter-swarm localization of a primary moves the entire swarm including those FLSs that serve as anchors for other swarms. These result in a high Hausdorff and Chamfer distances.

### 5.4 A Comparison with SwarMer

SwarMer [3] is a decentralized localization framework for FLSs. Individual FLSs localize relative to one another to form swarms.

---

[4]Depth decreases to 12 with $G$=50.

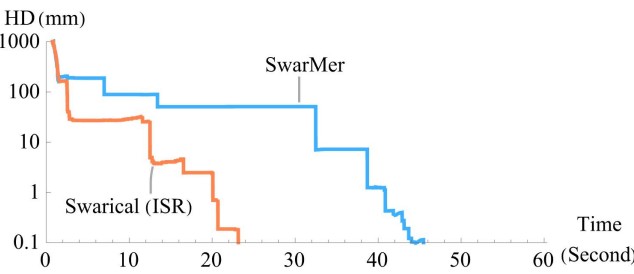

**Figure 15: Comparison of Swarical with SwarMer for the Skateboard. Click anonymized video links for a demonstration.**

An FLS of one swarm localizes relative to an anchor FLS of another swarm to merge with it, forming a larger swarm. This process repeats, causing swarms to merge until there is one swarm. Subsequently, SwarMer thaws the final swarm into individual FLSs and repeats the process again.

Both SwarMer and Swarical are continuous techniques that use the concept of localizing and anchor FLSs. SwarMer constructs its swarms in an online manner. Swarical is different because it constructs its swarms in an offline manner. SwarMer's swarms are seeded with 1 FLS that merge to construct larger swarms, ultimately growing into one swarm that includes all FLSs. Swarical's swarms are static. Swarical is an integrated approach that considers the range of sensors mounted on an FLS to track another FLS. This is reflected in its hierarchical swarm-tree and $nG$ FLS-trees. These concepts are absent from SwarMer.

Figure 15 shows the HD with SwarMer and Swarical for the Skateboard. Swarical is configured with group size 50 ($G$=50) and the ISR technique. SwarMer does not consider the error associated with the range of an FLS's tracking device. Hence, we assumes the tracking device is 100% accurate in measuring the pose of an FLS with both techniques. These results show Swarcial localizes the FLSs more than 2x faster than SwarMer. A similar observation is made with CD.

Swarical is faster than SwarMer for two reasons. First, FLSs exchange fewer messages. More specifically, SwarMer requires a challenge phase for a localizing FLS to identify its anchor FLS. This step is absent from Swarical; its Planner computes the localizing and anchor FLSs in an offline manner. Second, FLSs move a shorter distance with Swarical when compared with SwarMer. In the experiments of Figures 15, on the average 8% less. The minimum distance moved by FLSs with Swarical is 12% shorter than SwarMer.

## 6 CONCLUSIONS AND FUTURE RESEARCH

Swarical is a framework that considers the range of sensors mounted on FLSs to generate point clouds that enable FLSs to localize with a high accuracy. In turn, this renders highly accurate illuminations. The accuracy of Swarical is dictated by the sensor and its hardware used to localize. Swarical ensures localizing FLSs have line of sight with their anchors. We have simulation results showing Swarical is as accurate with scaled-down versions of drones, cameras, and ArUco markers. Our immediate research direction is to construct these candidate FLSs.

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
