# OpenReview forum: "Swarical: An Integrated Hierarchical Approach to Localizing Flying Light Specks"
_acmmm.org/ACMMM/2024/Conference — MM2024 Poster_

### Official Review · Reviewer_L3m3 · 2024-05-08

**Rating:** 6
**Confidence:** 3

**Summary:**

Swarical introduces a pioneering approach to localization for miniature drones known as Flying Light Specks (FLSs). Leveraging their hardware capabilities, it adeptly localizes and illuminates intricate shapes by converting mesh files into point clouds. Swarical's flexibility extends to accommodating various FLS orientations, ensuring robust performance. Compared to decentralized techniques, it excels with comparable accuracy while delivering twice the speed, making it a promising advancement in drone localization technology.

**Strengths:**

1. The proposed framework is quite novel and valuable for both academic and industrial community. As a new media display method, the reviewer thinks such a FLSs based framework has enough potential to be devoted by existing researchers.
2. The work is quite solid in both theoretical and engineering. Theoretically, the authors framework has achieved SOTA localization and tracking performance. On the engineering aspect, the Swarical framework offers a basic platform for subsequent researchers to also work in such a topic.

**Limitations:**

The reviewer is somewhat confused and want to know more about the FLS's future application scope, as a new type of 3D media display. Is this technology primarily suitable for exhibitions or artistic performances? Does it have broader application scopes? Besides, as a new type of 3D media display, I want to know how many FLSs are usually required to achieve such a display, since this is closely related to the cost. Is there a problem of high cost in such a media display way? I hope to hear the response from authors.

**Suitability:**

3

---

### Official Review · Reviewer_R2L9 · 2024-05-22

**Rating:** 2
**Confidence:** 2

**Summary:**

This paper presents the swarm-based hierarchical localization (Swarical) method for Flying Light Specks (FLSs), to localize and illuminate objects. It takes the range of sensors mounted on FLSs into account to generate high accuracy point clouds. Experimental results show the strong ability in FLS localization.

**Strengths:**

(1)A novel framework, namely Swarm-based hierarchical (Swarical), is proposed that considers hardware, software and data to localize FLS.
(2)A mixture of Swarm-tree and FLS-tree architecture is proposed to facilitate intra and inter swarm localization, which helps to improve the accuracy of positioning.
(3)Three different localization techniques for Swarical are introduced, including HC, ISR and RSF, and they are compared in the experimental section.
(4)From the experimental results, it can be seen that the positioning accuracy of Swarial is better than that of SwarMer.

**Limitations:**

(1)There are some issues with the template of the paper, where a 2018 template is used.
(2)Figure 2 in the paper did not explain its specific function or explain its composition clearly.
(3)The principle of the planner module is not well explained, and it is unclear how it processes mesh files, G, and Physical Spaces of FLSs and sensors to ultimately obtain Swarm tree and FLS tree.
(4)The structure arrangement of Section 4 is chaotic, and the three positioning techniques should be separated into a separate sub-section.
(5)The experimental section lacks analysis of ablation experiments for each module.

**Suitability:**

2

---

### Official Review · Reviewer_yYVP · 2024-06-09

**Rating:** 4
**Confidence:** 3

**Summary:**

This work introduces a framework called Swarical, which is projected to deal with miniature drones (FLSs) equipped with light sources that are used to create 2D and 3D multimedia displays. The main goal of this suggested framework is to improve the accuracy of localizing these drones, resulting in a more accurate multimedia presentation. It can be argued that at the framework's core is the Planner, which is responsible for transforming the mesh files into point clouds of drones, building hierarchical trees for the localization of FLSs, and managing the distances between them.

The study also presents a proof of concept by applying the framework with Raspberry Pi cameras and markers to monitor the position and orientation of the FLSs. In general, the research shows that proposed framework Swarical is more than twice as rapid as cutting-edge techniques while maintaining similar accuracy, and it seems robust for infrared interventions.

**Strengths:**

One positive aspect of the approach proposed by the authors is the Swarical's ability to track the FLS. According to the authors, the framework can achieve a millimeter-level precision in terms of position and less than a one-degree error in orientation.

Another highlight of their work is the improvement in communication overhead and energy consumption due to reducing the message exchanges and movement distances among the FLS. The results show that the framework is also more than twice as fast as the state of art concurrent technique, SwarMer,

The authors have open-sourced their software implementations and datasets, which is a significant positive aspect as it allows collaboration and validation rounds to be conducted by the community.

**Limitations:**

The results are promising, particularly those related to the performance of the proposed framework compared to the state-of-the-art alternative. However, despite the promising aspects, the authors need to address specific points to enhance the quality of the manuscript: The top priority is the organization and presentation of the study!


1) **Clarity**: In any scientific paper, the adopted methodology must be detailed enough to be replicable by the reader. The approach assumed by the authors needs to clarify the research flow. As it stands, it is challenging to determine what the methods/methodology is and what the results are. Additionally, the authors mix several topics in a single paragraph. For example, in the text below, the authors address the relationship between the camera and markers, the content of Figure 6, the content of Table 1, and the content of Figure 7. It is confusing for the reader to assimilate the content as the authors do not delve into the discussion.
     - "The maximum range of each camera for detecting a marker depends on the size of the marker. Figure 6 shows the detection rate with a 4.7 mm paper printed maker size. While the x-axis of this figure is the distance between the camera and the marker, the yaxis is the detection rate. It highlights the minimum focus range of Table 1 with the detection rate becoming 100% at those minimums. With the wide angle lens, the detection rate drops to zero with 300mm. With this marker size, Figure 7 shows the percentage error increases as a function of the distance2. The regular lens provides a lower error as a function of longer distances when compared with the wide lens."

2) **Robustness of Tests**: The authors mention that the proposed method is up to twice as fast as the main alternative, but the details of the tests still need to be discussed. Some questions that need to be answered and can impact the described results:
   - Was the test environment controlled? If so, what variables were controlled? Light control? What are the expected behaviors for variables such as changes in lighting or wind?

3) **Communication Failure**: The authors do not adequately mention the behavior of the framework if one or more FLSs fail during operation.
Is there any recovery mechanism?

4) **Scalability**: The authors mention that the framework can adjust to the number of FLSs but do not detail the limits or the quantity used in the tests.

**Suitability:**

3

---

### Meta-Review · Area_Chair_gEMd · 2024-06-25

**Recommendation:** Accept (Poster)
**Confidence:** 5

**Metareview:**

The paper presents Swarical, a hierarchical localization technique for Flying Light Specks (FLSs). The reviewers noted the novelty and solid engineering of the proposed framework, as well as its potential impact on the multimedia community. The concerns raised regarding clarity, test robustness, and scalability have been satisfactorily addressed in the authors' rebuttal. Therefore, I recommend accepting this submission.